# Chimeric design of pyrrolysyl-tRNA synthetase/tRNA pairs and canonical synthetase/tRNA pairs for genetic code expansion

Wenlong Ding[1,3], Hongxia Zhao[1,3], Yulin Chen[1], Bin Zhang[1], Yang Yang[2], Jia Zang[1], Jing Wu[1] & Shixian Lin [1✉]

An orthogonal aminoacyl-tRNA synthetase/tRNA pair is a crucial prerequisite for site-specific incorporation of unnatural amino acids. Due to its high codon suppression efficiency and full orthogonality, the pyrrolysyl-tRNA synthetase/pyrrolysyl-tRNA pair is currently the ideal system for genetic code expansion in both eukaryotes and prokaryotes. There is a pressing need to discover or engineer other fully orthogonal translation systems. Here, through rational chimera design by transplanting the key orthogonal components from the pyrrolysine system, we create multiple chimeric tRNA synthetase/chimeric tRNA pairs, including chimera histidine, phenylalanine, and alanine systems. We further show that these engineered chimeric systems are orthogonal and highly efficient with comparable flexibility to the pyrrolysine system. Besides, the chimera phenylalanine system can incorporate a group of phenylalanine, tyrosine, and tryptophan analogues efficiently in both *E. coli* and mammalian cells. These aromatic amino acids analogous exhibit unique properties and characteristics, including fluorescence, post-translation modification.

[1] Life Sciences Institute, Zhejiang University, Hangzhou 310058, China. [2] School of Chemistry and Chemical Engineering, Nanjing University, Nanjing 210046, China. [3]These authors contributed equally: Wenlong Ding, Hongxia Zhao. ✉email: sxlin@zju.edu.cn

Genetic code expansion strategy allows site-specific incorporation of unnatural amino acids (UAAs) into target proteins. The installed UAAs carry a variety of chemical functionalities that are not present on the natural amino acid side chains, which can be used as a handle for protein function studies[1–3]. In this context, the incorporation of UAAs has enabled a broad range of fundamental and applied advances. These applications include imaging protein localisation[1,3–6], synthesis of antibody-drug conjugates[3,7–9], decoding the effect of post-translational modifications[10–12], capturing protein–protein interactions[13–16], manipulation of protein function[17–20] and engineering enzymes with novel functions[21–23].

A crucial prerequisite for genetic code expansion is an orthogonal aminoacyl-tRNA synthetase (aaRS)/tRNA pair, which must be orthogonal to all endogenous aaRS/tRNA pairs in the host species[1–3,24–26]. Several aaRS/tRNA pairs that are orthogonal to either prokaryotic or eukaryotic cells (or called partially orthogonal pairs) have been developed successfully and utilised broadly over the past two decades. Despite the tremendous success, most partially orthogonal pairs are limited in many ways[1,26–29]. For example, in order to encode a desirable UAA in prokaryotes and eukaryotes, two distinct pairs and separate protein evolution procedures in *E. coli* and yeast are required, which is inefficient and laborious. In this case, engineering orthogonalized platforms through functionally replacing the endogenous translation system in *E. coli* for genetic code expansion in *E. coli* and mammalian cells is a remarkable strategy. This strategy has been successfully applied to the tyrosine[30] and tryptophan[28] translation systems. However, due to the dependence on partially orthogonal pairs, this strategy is only applicable in the engineered *E. coli* host rather than other prokaryotes. In contrast, the pyrrolysyl-tRNA synthetase (pylRS)/pyrrolysyl-tRNA (pylT) pairs, originated from *Methanosarcina mazei* (*Mm*) and *Methanosarcina barkeri* (*Mb*) respectively, are discovered to be fully orthogonal in a wide range of prokaryotes and eukaryotes. The pylRS/pylT pair is evolved in Archaea to decode amber stop codon with pyrrolysine, the twenty-second amino acid[31,32]. With this unique system, genetic code expansion of eukaryotes can easily rely on the pylRS/pylT pair that is engineered in the prokaryotic selection system (e.g., *E. coli*). This facile nature of the pyrrolysine system has profoundly advanced our ability to engineer diverse UAAs incorporation systems for various model organisms[1,25–27,29,33]. The pylRS/pylT pair is thus commonly considered as an ideal pair for genetic code expansion in both prokaryotes and eukaryotes. Unfortunately, other flexible and efficient systems are still largely unknown. Therefore, the discovery of other fully orthogonal pairs with the comparable flexibility (e.g., broad orthogonality and high efficiency) to the pylRS/pylT system is extremely valuable and under high demand for advancing the process of incorporating UAAs with diverse structural scaffolds and distinct functions[25,26].

We seek to exploit the orthogonality of the pyrrolysine system to create a panel of orthogonal aaRS/tRNA pairs with the comparable flexibility in their applications. *Mm* or *Mb* pylRS/pylT pair uses a unique mechanism to achieve their orthogonality to all endogenous aaRS and tRNA[34–37]. Firstly, pylT has a highly unusual structure, lacking many of the conserved invariant structural elements that are present in canonical tRNAs. Among all the unusual structural features, the short variable region, the nonstandard T-loop and the small D-loop contribute to the orthogonality of pylT (highlighted by the dashed box in Fig. 1)[36,38,39]. Secondly, the tRNA binding domain (TD) of pylRS is folded into a compact globule structure, which is not observed in other aaRSs. The pylRS TD binds to the pylT exclusively, through recognition of its unusual structures, which contributes to the orthogonality of pylRS[37–39]. Thus, the directed evolution of the pylRS-TD significantly improves the activity of the pyrrolysine system[37,40]. Finally, most aaRSs bind to

the cognate tRNAs in their anticodon bases to ensure fidelity of the aminoacylation process, while the pylRS does not directly bind the anticodon of pylT, which further facilitates the decoding of distinct sense codons through mutating the corresponding anticodons[41,42] (Fig. 1). Together, the combination of unique structural features present in both the pylT and pylRS results in the orthogonality and activity of the pyrrolysine system. We, therefore, envision that orthogonal aaRS/tRNA pairs carrying these unique structural features may have the potential of broad orthogonality and activity as the pyrrolysine system for genetic code expansion.

Here, we report the chimera design for the engineered orthogonal aaRS/tRNA pairs by rationally transplanting the key orthogonal components from the pyrrolysine system into several aaRS/tRNA pairs of choice (Fig. 1). The resulting chimeric tRNA synthetase/chimeric tRNA pairs can be further engineered to be fully orthogonal and highly active for suppressing of their corresponding amino acids in a site-specific manner. Besides, by engineering in the amino acid binding pocket, the chimeric phenylalanine system is repurposed to site-specifically incorporate Phe, Tyr and Trp analogues (Supplementary Fig. 1) in prokaryotes and eukaryotes. For several Phe analogues, higher incorporation efficiency is achieved with the engineered chimeric PheRS relative to other systems. Among these UAAs, an amino acid with a post-translational modification (L-3,4-Dihydroxy-Phe) and a set of N1-, C6-, C7- substituted Trp analogues are incorporated into mammalian cells. (Supplementary Fig. 1) Site-specific installation of these UAAs is demonstrated with an *E. coli* to mammalian shuttle system. Furthermore, we have shown that the successful incorporation of fluorescent amino acid, 6 or 7-Cyano-Trp. These amino acids allow two-photon imaging of protein of interest in mammalian cells with minimum structural perturbation due to its relatively high brightness and small size. Our chimera design may serve as a general strategy with additional directed protein and/or tRNA engineering to orthogonalize any given tRNA synthetase/tRNA pair to incorporate UAAs with distinct chemical functionalities into protein.

## Results

**Engineering a chimeric histidyl-tRNA**. To engineer chimeric aaRS/tRNA pairs with the comparable flexibility to the pyrrolysine system, key orthogonal components from the pylRS and the pylT were chosen for rational chimera fusion. (Fig. 1) As a proof-of-principle, we first attempted to orthogonalize *E. coli* histidyl-tRNA synthetase (*Ec*.hisRS; *Ec* is omitted hereafter)/histidyl-tRNA (*Ec*.hisT; *Ec* is omitted hereafter) pair. HisRS has two separate domains, a CD and a TD, which are joined by a short peptide linker. The clear distinction between the CD and the TD in the hisRS facilitates the design of a fusion protein between the pylRS TD and the hisRS CD without affecting the catalytic activities of the latter[43]. HisT has an uncommon G^−1 base, which is critical for its binding to the catalytic domain of hisRS[43,44] (Fig. 2a and Supplementary Fig. 2A). These features of hisRS and hisT help simplifying our chimera design.

With hisRS and hisT candidates in hand, we then set out to engineer an orthogonal chimera hisT using pylT as a template, with the focus on the acceptor arm (Fig. 2a and Supplementary Fig. 2A). First, a series of chimera hisTs (chHisTs) was generated by swapping the pylT acceptor arm with serial truncations of the acceptor arm sequence from the hisT (Fig. 2b and Supplementary Fig. 2B). Next, the chimera hisRS (chHisRS-1) was constructed by fusing the pylRS TD with the hisRS CD. (Fig. 2c) Subsequently, a series of constructs with different chHisT designs along with the chHisRS-1 was built to investigate the amber suppression efficiency. In these constructs, expression of chHisRS was driven

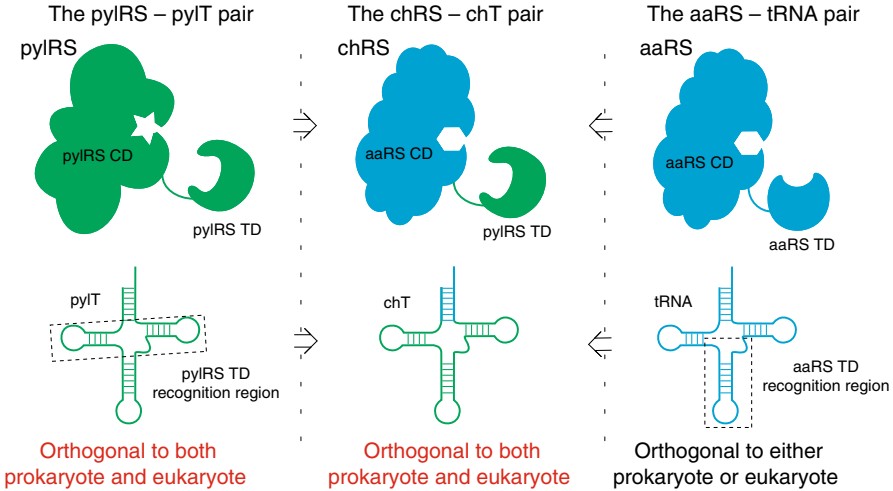

**Fig. 1 The chimera design for orthogonalizing aaRS/tRNA pairs. a** Cartoon structures of the pylRS/pylT pair (left panel); Cartoon structures of a common aaRS/tRNA pair (right panel); Cartoon structures of a chimeric aaRS (chRS) by fusing pylRS tRNA binding domain (TD) and aaRS catalytic domain (CD), a chimeric tRNA (chT) by replacing the pylT acceptor arm with an acceptor arm of a given tRNA (middle panel). The regions on the tRNAs that binds to the aaRS TD are highlighted by dashed boxes.

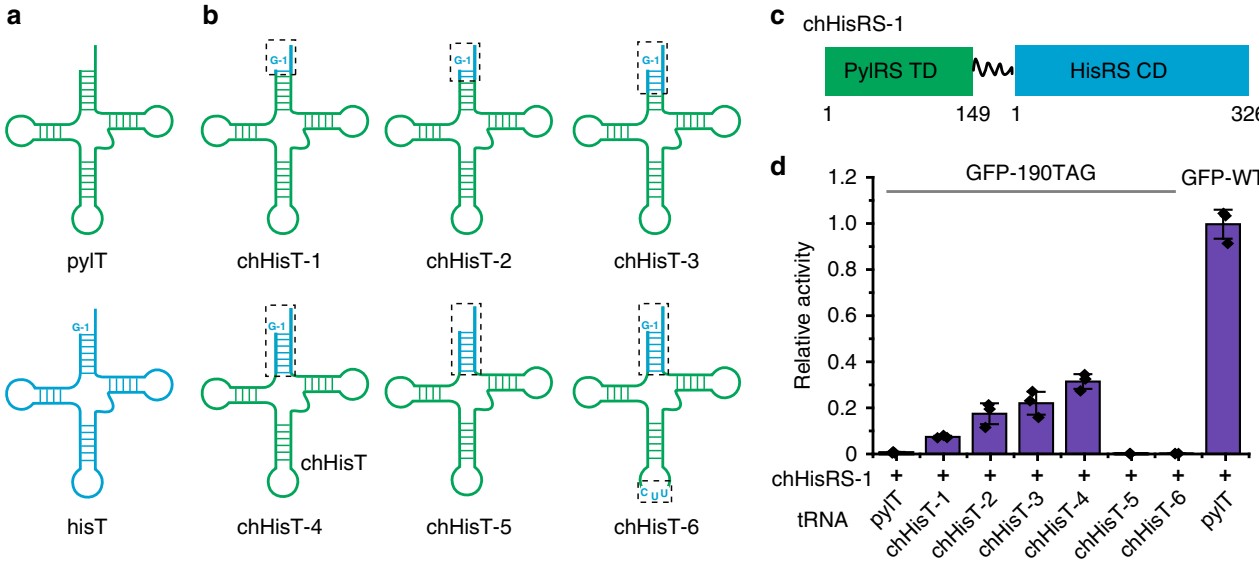

**Fig. 2 Identifying a chimeric histidyl-tRNA. a** The cloverleaf structures of pylT and hisT with a $G^{-1}$ structure. The detailed sequence information of these tRNAs is shown in Supplementary Fig. 2. **b** The cloverleaf structures of chimeric histidyl-tRNAs (chHisTs) in which the sequence from pylT is coloured in green, while the sequence from hisT is coloured in blue and highlighted by a dashed box. The detailed sequence information of chHisTs is shown in Supplementary Fig. 2 as well. **c** Cartoon structure of chimeric histidyl-tRNA synthetase-1 (chHisRS-1) by fusion of pylRS TD (residues 1-149) and hisRS CD (residues 1-326). The truncation residue number is shown under the cartoon. **d** Amber suppression efficiency is tested by a GFP reporter assay. Wild-type GFP is expressed under the same condition as the control. The fluorescent intensity in each group is measured by a plate reader and normalised by the wild-type. Error bars represented ±standard error of the mean($n = 3$). Source data are available in the Source Data file.

by the constitutive, mild-strength *E. coli* glutaminyl-tRNA synthetase (*glnS*) promoter; and expression of chHisT was controlled by the *E. coli lpp* gene promoter. Amber suppression ability of this chimera system was assayed by monitoring the expression of a full-length GFP carrying the 190TAG mutation. The results showed that chHisT-4 carrying the entire acceptor arm from hisT exhibited the highest amber suppression efficiency, which achieved 35% GFP expression compared with wild-type GFP under the same expression conditions. Notably, the amber suppression efficiency of chHisT-4 is much higher than chHisT-1 to -3. ChHisT −1 to −3 carry a shorter acceptor arm from hisT relative to chHisT-4, which may weaken the binding affinity to chHisRS-1. In contrast, no GFP expression was

detected in the group expressing pylT, chHisT-4 without the $G^{-1}$ residue (chHisT-5) or chHisT-4 with CUU as the anticodon (chHisT-6) (Fig. 2b, d and Supplementary Fig. 2B). These data demonstrated that an efficient chHisT can be engineered with the chimera strategy, and the chHisRS/chHisT pairs are active.

**Engineering a chimeric histidyl-tRNA synthetase.** To further increase the flexibility of our chimeric histidine system, we then turned to engineer the chHisRS. We designed a series of chHisRSs (from −1 to −5) by fusing the pylRS TD or its mutated forms (IPYE mutants)[40] with the hisRS CD at either the N- or C-terminus (Fig. 3a). The amber suppression efficiency of these chHisRSs with the most active chHisT-4 (chHisT-4 was referred

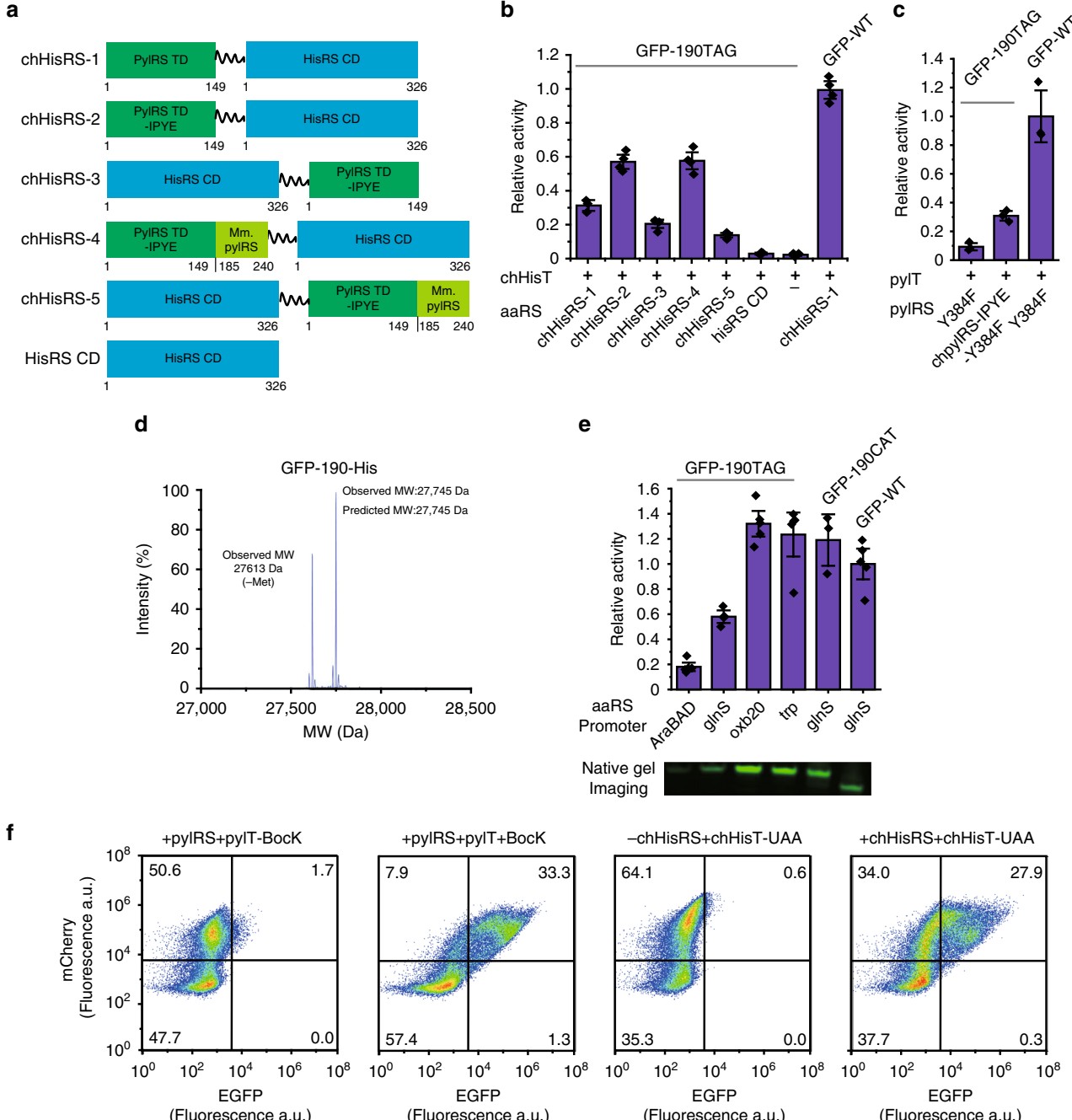

**Fig. 3 Identifying a chimeric histidyl-tRNA synthetase. a** Cartoon structures of chimeric histidyl-tRNA synthetases (chHisRSs) in which the sequence from pylRS is coloured in green and the sequence from hisRS is coloured in blue. The pylRS TD mutant form carries IPYE mutations. ChHisRS-4 and chHisRS-5 carry a chimeric pylRS TD comprising residues 1-149 from *Mb*.pylRS and 185-240 from *Mm*.pylRS, which is reported to promote amber suppression efficiency[40]. **b** Analysis of amber suppression activity of the generated chHisRSs by GFP reporter assay at 30 °C ($n = 4$). **c** Analysis of amber suppression activity of the generated pylRSs by GFP reporter assay in the presence of 2 mM Boc-L-lysine (BocK) ($n = 3$). **d** Mass spectrometry characterises the fidelity of His incorporation into GFP. **e** Analysis of amber suppression activity of the chHisRS under different promoters by GFP reporter assay and a native in-gel fluorescence imaging assay. (Supplementary Fig. 3D) GFP-190CAT (GFP-190His) and GFP-WT (GFP-190Asp) are used as controls ($n = 3$ for GFP-190ACT, $n = 4$ for GFP-190TAG *glnS* promoter and $n = 5$ for others). **f** Flow cytometry analysis of amber suppression efficiency in mammalian cells transfected with the indicated orthogonal systems. For the pylRS system, BocK (2 mM) is added to the medium of HEK 293T cells. For the chHisRS system, the addition of UAA is not necessary, and His in the medium is used as the substrate. Error bars represented ±standard error of the mean. Source data are available in the Source Data file.

as the chHisT hereafter) was tested. We detected nearly 60% GFP expression compared with the wild-type GFP in the presence of either the chHisRS-2 or chHisRS-4 with the chHisT. Both chHisRS-2 and chHisRS-4 contained the pylRS TD with IPYE mutants. This observation is consistent with the previous report that IPYE mutations on the pylRS TD improved amber suppression efficiency of the pyrrolysine system[37,40]. Generally, chHisRSs with pylRS TD at the N-terminus fusions showed a more significant improvement of amber suppression efficiency compared with the corresponding C-terminus fusions. Notably,

we detected very low amber suppression with the expression of hisRS CD or chHisT alone, suggesting the excellent orthogonality of chHisT in vivo and the importance of pylRS TD in the chimeric synthetase (Fig. 3b). Impressively, under the same expression conditions, our chHisRS/chHisT pair showed higher His incorporation efficiency than the previously engineered pyrrolysine system[40] with 2 mM of Boc-L-Lysine, (Fig. 3b, c) testifying the high efficiency of the chimeras. Furthermore, the chimera histidine pair can incorporate His with extremely high fidelity, which was confirmed by LC-MS (Fig. 3d and Supplementary Fig. 3A, B) and LC-MS/MS analysis (Supplementary Fig. 3C). Surprisingly, when chHisRS expression was driven by a stronger promoter such as oxb20 or trp, we observed similar levels of GFP between the wild-type GFP-190ACT (GFP-190His) and the chimeric histidine system. (Fig. 3e) Indeed, our chimeric system produced very little truncated protein with stronger synthetase promoters (Supplementary Fig. 3E). We also tested the amber suppression efficiency of this system in another model protein, ubiquitin. The yield of purified ubiquitin protein was used to assay the robustness of amber suppression efficiency for our chimeric systems (Supplementary Fig. 4). Together, these results demonstrate that this generated chHisRS/chHisT pair is a promising system in the field of genetic code expansion.

To gain insight into the structural basis of our chimeric histidine system, we generated a protein complex model using PyMOL by super-positioning relevant crystal structures with both pylRS TD and hisRS CD[36,37,43]. The phosphorus atoms of the tRNAs in these structures were used for the superposition alignment in the PyMOL programme (Supplementary Fig. 5). The superposition structure indicated the C-terminus of the pylRS TD and the N-terminus of the hisRS CD are juxtaposed to each other within 0.5 nm (Supplementary Fig. 5), which may be critical for the high activity of chHisRS towards its cognate chHisT. We also envisioned that the linker-length between the TD and CD in the chHisRS might be important for chHisT binding and turnover during aminoacylation (Supplementary Fig. 5). Indeed, the chHisRS fused with an eighteen GlySer (GS)-rich amino acid linker is slightly more efficient, showing >60% GFP expression to wild-type GFP (Supplementary Fig. 6). Finally, in our chimera design, chHisRS and chHisT bind to each other in an anticodon bases independent manner. Thus, this system enables the decoding of diverse codons (e.g., all three stop-codons) through mutation of corresponding anticodons without abolishing its activity (Supplementary Fig. 7). These results demonstrate that the chimeric histidine system we engineered is highly active and that the chHisT is orthogonal in prokaryotic cells.

**The orthogonality of chimeric histidine system**. Next, we investigated the function of our chimeric histidine system in eukaryotic cells and the orthogonality of our chimeric histidine system in vitro. We co-transfected a plasmid carrying chHisT and chHisRS genes, and a plasmid harbouring mCherry-T2A-EGFP (190TAG) into mammalian HEK 293T cells. Expression of full-length GFP was monitored by fluorescence-activated cell sorting (FACS) and used as readout to access the efficiency of our chimeric histidine system, while the expression of mCherry was used as a transfection control. Full-length GFP expression was observed clearly in the presence of both chHisT and chHisRS at a level comparable to that of the pyrrolsyine system in incorporating BocK[40]. In contrast, no detectable GFP expression was detected when the expression of chHisRS was omitted, suggesting that the chHisT is orthogonal in mammalian cells (Fig. 3f and Supplementary Fig. 8). Meanwhile, we found that the chHisT gene with a CCA tail showed nearly fivefolds increase in activity

compared with the gene without a CCA tail (Supplementary Fig. 9). To further confirm the orthogonality of chHisRS, we performed an in vitro aminoacylation experiment and tested the orthogonality of chHisRS with all endogenous tRNAs from E. coli cells and mammalian cells, using the E. coli hisRS as a positive control and its catalytic domain alone as a negative control. The aminoacylation activity of chHisRS was significantly higher in the group with the addition of chHisT. In contrast, background activity was detected in the group without adding chHisT or with the catalytic domain alone (Supplementary Fig. 10A). We also tested the orthogonality of chHisRS to hisTs from both E. coli and mammalian cells with the CUA anticodon (Supplementary Fig. 2C) in E. coli. No GFP signal was detected in the presence of chHisRS using these tRNAs, further supporting the orthogonality of chHisRS in vivo (Supplementary Fig. 10B). Together, these data testified that our chimeric histidine system is highly orthogonal and can achieve high incorporation efficiency in both prokaryotic and eukaryotic cells.

**Engineering additional orthogonal pairs**. Next, we asked if other aaRS/tRNA pairs could be orthogonalized with this rational chimera design. The main challenge in extending our design is to orthogonalize aaRSs. Unlike hisRS, the catalytic domain and the tRNA binding domain in most canonical aaRSs are usually fused, in contrast to the clear separation between those two domains in hisRS. Therefore, it is difficult to predict whether the truncated version of its catalytic domain will affect the catalytic activity or not and whether the fusion proteins allow proper tRNA binding and aminoacylation. We surveyed the available structures of various aaRSs from E. coli or human in the Protein Data Bank and aimed at isolating its catalytic domain with minimum influence on its aminoacylation activity. We chose 16 additional aaRSs as candidates for our engineering pipeline. Through the rational design of these aaRSs by fusing the pylRS TD at the N-terminus of the catalytic domain and engineering its cognate tRNA based on the experience of the chHisT, we successfully identified seven active chimeric aaRS/tRNA pairs (Fig. 4a and Supplementary Fig. 11). Although the protein expression level of the catalytic domains was much higher than the full-length chimeric synthetases (Supplementary Fig. 12), only background activity was detected when the catalytic domain was used, or when chimeric tRNA was present alone (Fig. 4a). Interestingly, we were able to engineer the chimeric phenylalanyl-tRNA synthetase (chPheRS) from human mitochondrial PheRS with about 6% of protein expression compared to wild-type GFP, in contrast to <0.5% of protein expression when using highly similar PheRS from E. coli (Fig. 4a and Supplementary Fig. 11). This observation indicated that subtle changes in the chimeric aaRSs may result in completely different activities of chimeric enzymes. Although these chimeras were active, the amber suppression efficiency of these systems was still very low. In order to improve the efficiency, the chimeric synthetases were further engineered in terms of its truncation position and linker length, as well as optimising the protein expression through promoter survey. In chPheRS, we observed that a strong promoter was the major contributor to the efficiency improvement from 6 to 50% (Fig. 4b and Supplementary Fig. 13A, B), which is likely due to the low expression level of full-length chimeric synthetases (Supplementary Fig. 12). In comparison, no amber suppression was detected in the presence of chPheT alone, demonstrating the excellent orthogonality of chPheT (Fig. 4b and Supplementary Fig. 13A). Also, chPheRS showed good orthogonality to endogenous tRNAs from both E. coli and mammalian cells by in vitro aminoacylation experiments with the catalytic domain as a negative control and the PheRS as a positive control (Fig. 4d and Supplementary Fig. 14A). Similarly,

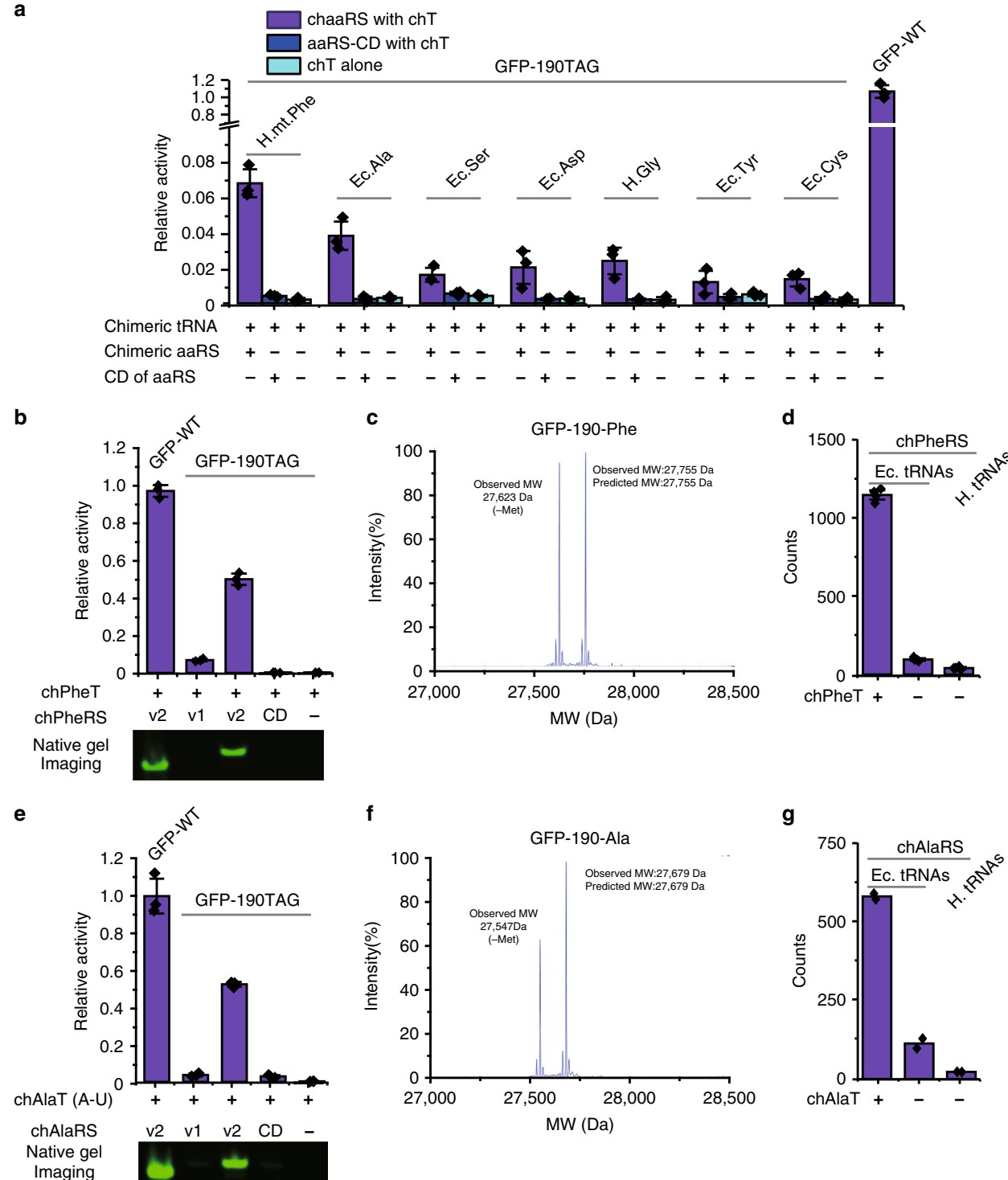

we observed extremely high incorporation fidelity of Phe by LC-MS (Fig. 4c and Supplementary Fig. 13C, D) and LC-MS/MS analysis (Fig. Supplementary Fig. 13E).

Furthermore, we observed that chimeric alanyl-tRNA synthetase (chAlaRS) had a very high activity towards its corresponding chimeric alanyl-tRNA (chAlaT), with over 70% amber suppression efficiency when the aaRS expression was driven by the *glnS* promoter. Unfortunately, the engineered chAlaT was not orthogonal to the endogenous alaRS in *E. coli*, which predominantly recognised the unusual G–U pair on the receptor arm[45] (Supplementary Fig. 15A, B). To improve the orthogonality, we

attempted to use a chAlaT with A–U pair instead of G–U pair (Supplementary Fig. 2D). Indeed, by removing the G–U pair recognition loop in the chAlaRS, together with synthetase promoter optimisation, we generated chAlaRSs that can efficiently suppress chAlaT with A–U pair. The chAlaT with A–U pair was named as the chAlaT hereafter. With this chAlaRS/chAlaT pair, nearly 50% of protein expression was achieved compared to the wild-type GFP (Fig. 4e and Supplementary Fig. 15C, D). No amber suppression was detected in the presence of chAlaT alone (Fig. 4e and Supplementary Fig. 15C), demonstrating the excellent orthogonality of the chAlaT. In

**Fig. 4 Engineering additional orthogonal aaRS/tRNA pairs with the chimera design. a** Analysis of amber suppression activity of generated seven chimeric aaRSs towards their corresponding chimeric tRNAs by GFP reporter assay using the corresponding catalytic domain and chimeric tRNA alone as controls ($n = 3$). In the figure, $Ec$ represents *E. coli*, $H$ means human, and $H.mt$ stands for human mitochondria. Orthogonal pairs are labelled with their amino acid three-letter abbreviations. And protein expression was driven by the *glnS* promoter. **b** Analysis of amber suppression activity of the generated chPheRSs by GFP reporter assay and a native in-gel fluorescence imaging assay ($n = 3$). In the figure, chPheRS v1 and v2 were chPheRS that were driven by *glnS* and *oxb20* promoter, respectively (Supplementary Fig. 13). **c** Mass spectrometry characterisation of the fidelity of Phe incorporation into GFP with chPheRS v2. **d** Purified chPheRS is incubated with tRNAs extracted from *E. coli* cells (w/o the expression of chPheT) or mammalian cells in the presence of radiolabelled Phe. The aminoacylation reactions are determined by measuring the production of radioactive aminoacyl-tRNA with a liquid scintillation counter ($n = 4$). (Supplementary Fig. 14A) **e** Analysis of amber suppression activity of the generated chAlaRSs by GFP reporter assay and a native in-gel fluorescence imaging assay ($n = 3$). In the figure, chAlaRS v1 and v2 were chAlaRS that were driven by *glnS* and *oxb20* promoter, respectively (Supplementary Fig. 15). **f** Mass spectrometry characterisation of the fidelity of Ala incorporation into GFP with chAlaRS v2. **g** Purified chAlaRS is incubated with tRNAs extracted from *E. coli* cells (w/o the expression of chAlaT) or mammalian cells in the presence of radiolabelled Ala. The aminoacylation reactions are determined by measuring the production of radioactive aminoacyl-tRNA with a liquid scintillation counter ($n = 2$). (Supplementary Fig. 14B) Error bars represented ±standard error of the mean. Source data are available in the Source Data file.

addition, high orthogonality of chAlaRS was observed by the in vitro aminoacylation assay (Fig. 4g and Supplementary Fig. 14B), and high incorporation fidelity was confirmed by LC-MS (Fig. 4f and Supplementary Fig. 15E, F) and LC-MS/MS analysis (Supplementary Fig. 15G). These data testify that our chimera strategy can be applied to multiple aaRS/tRNA pairs for its orthogonalization. Furthermore, the suppression efficiency of the chimeric aaRS can be dramatically improved towards its cognate tRNA through systematic engineering and expression optimisation of chimeric aaRS.

We noticed that all the chimeric tRNAs we introduced had the same structure at the bottom part of the cloverleaf structure from pylT, with the only difference between these tRNAs in their acceptor arms (Supplementary Fig. 2). We thus hypothesised that targeting the acceptor arms of these chimeric tRNAs for optimisation might further improve their binding affinity to their cognate chimera synthetases or accelerate the turnover rate of the chimeric tRNAs during aminoacylation process, and thereby, resulting in an increased amber suppression efficiency. Supporting this notion, in the superposition structure (Supplementary Fig. 5), we noticed that the first a few base pairs in the acceptor stem was the primary binding sites for the catalytic domain of the chimeric synthetases. We chose the chimeric serine system as the test for the hypothesis because previous efforts in linker and promoter optimisation led to little improvement (from 1.5 to 4.0%) for this system (Supplementary Fig. 16). A library of single base-pair mutation on the tRNA acceptor stem was generated. And through screening this library, we successfully identified a chSerT (chSerT-2) that significantly improved the amber suppression efficiency. The efficiency of the best chimeric serine system with chSerT-2 (nearly 15% of protein expression compared to wild-type GFP) was tenfold higher than the original one with extremely high fidelity (Supplementary Figs. 2E, 16). These results indicated that the combination of our chimera design and the tRNA evolution strategy may serve as a general platform to orthogonalize a given tRNA synthetase/tRNA pair, which broadly expands the scope for our orthogonalizing design.

**Incorporating unnatural amino acids**. Upon successfully engineering several orthogonal aaRS/tRNA pairs, we then tested whether the chimeras can be further engineered to alter its substrates specificity for the incorporation of UAAs beyond natural amino acids incorporation. The chimeric PheRS/tRNA pair was chosen as an example because of the structural similarity of mitochondrial PheRS to yeast PheRS, and the latter one has been applied to the genetic code expansion in *E. coli*[46–48]. In the chPheRS residues, Thr 467 and Ala 507 critical for exclusive Phe recognition were mutated to Gly to preclude the recognition of

endogenous Phe (Fig. 5a). Amber suppression efficiency of the synthetase variants to the Phe analogue, 4-Azido-Phe, was then tested. The results suggested that the chPheRS carrying both T467G and A507G mutations (the double mutant or chPheRS-1) can efficiently distinguish the 4-Azido-Phe from all endogenous amino acids (Fig. 5b). In contrast, synthetases bearing a single mutation primarily recognised endogenous amino acids (Fig. 5b). The incorporation of 4-Azido-Phe by chPheRS-1 was further verified by the in-gel fluorescent labelling of an alkyne containing fluorophore[49] with the purified protein (Fig. 5c), with the cell lysate (Supplementary Fig. 17B) and by LC-MS analysis (Supplementary Fig. 17C). Interestingly, chPheRS-1 showed a poly-specificity for other Phe analogues with structural similarities, such as 2-Naphthyl-Ala and 3-Cyano-Phe (Fig. 5d and Supplementary Fig. 17D, E). Since these UAAs have been incorporated by PylRS mutants[50,51] and *Mj*.TyrRS mutants[52,53], we used the same UAA in all three systems for direct comparisons of the amber suppression efficiency. Impressively, for the three Phe analogues we tested, the chPheRS-1 showed higher activity in site-specific UAAs installation than other systems (Fig. 5e, f). In addition, UAA incorporation efficiency with different systems was also assessed by the yield of purified protein (Fig. 5e, f and Supplementary Table 1), the results of which were highly consistent with those of GFP reporter assay. Furthermore, the specificity of evolved chPheRS was directly transferable to mammalian cells without further engineering, as expected, based on the broad orthogonality of the chimeric system in *E. coli* and mammalian cells. (Fig. 5g and Supplementary Fig. 18) Together, we demonstrated that the chimeric phenylalanine system is another *E. coli*–mammalian shuttle system for efficient genetic code expansion in both prokaryotes and eukaryotes, similar to the pyrrolysine system[10,54].

With the chimeric shuttle system in hand, we then tested whether the chPheRS can be further engineered to install amino acids with different frameworks (e.g., a tyrosine analogue), so as to expand our tool kits for protein studies. Tyrosine hydroxylation was recently identified as a histone post-translational modification with unknown biological function[55]. Besides, L-3,4-Dihydroxy-Phe (L-Dopa) can be easily oxidised and converted to a reactive dopaquinone intermediate on protein for subsequent bioconjugation[56,57]. However, the genetic code expansion of L-Dopa in mammalian cells remains elusive. We attempted to incorporate L-Dopa in mammalian cells with our chimeric phenylalanine system. First, L-Dopa was displayed into the crystal structure of PheRS by molecular docking. Based on the docked structure, residues (Q356, T467 and A507) that are surrounding the hydroxyl groups on L-Dopa were randomised and saturatedly mutagenized to generate a library of synthetase variants for the subsequent positive and negative selection, as

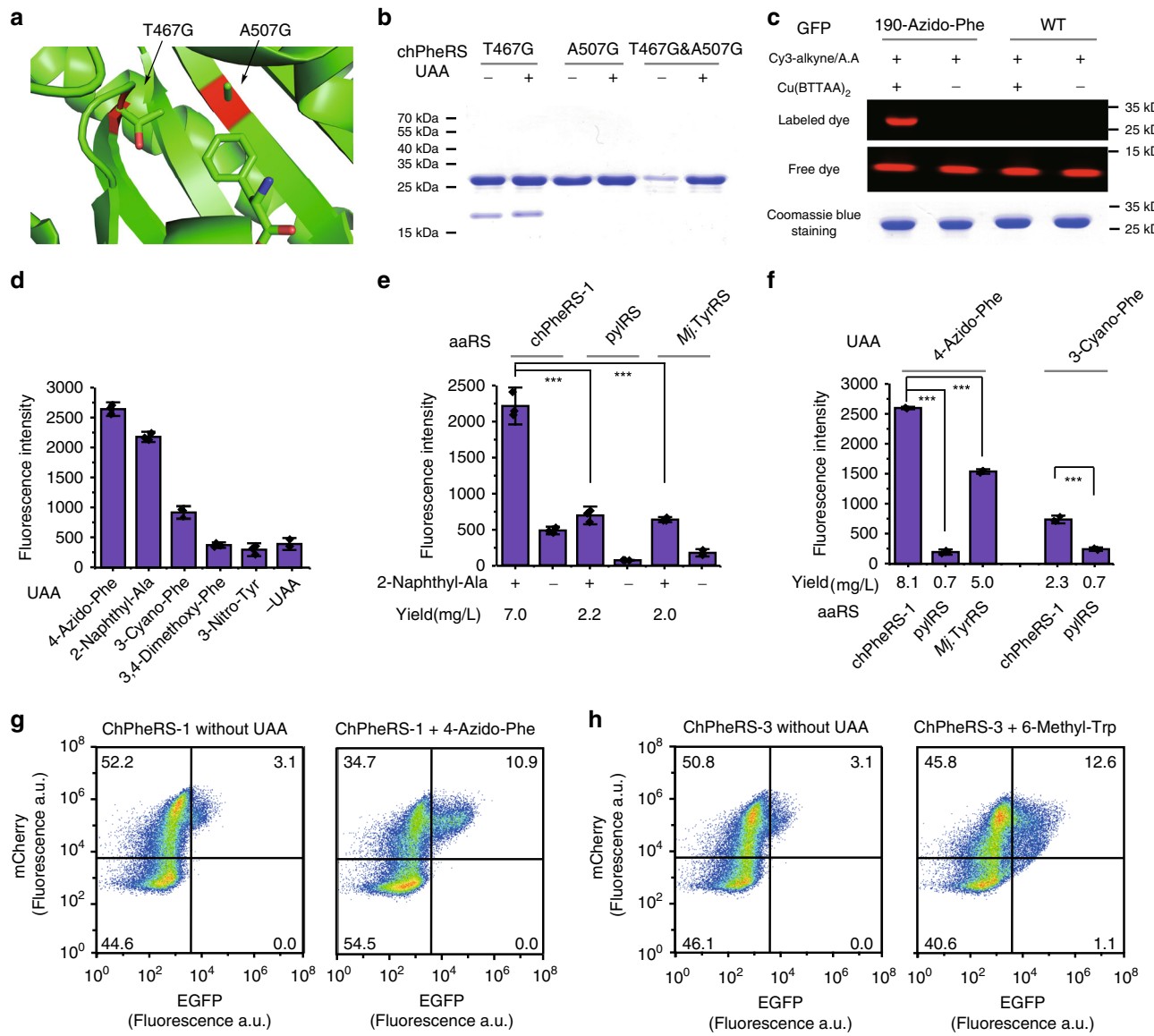

**Fig. 5 Incorporation of Phe and Tyr analogues with the chimeric PheRS/tRNA pair. a** Structure of human mitochondrial PheRS amino acid binding pocket, in which two gate-keeper residues are highlighted in red and shown sticks. **b** Analysis of amber suppression activity of the chPheRS variants to 4-Azido-Phe by detecting expression of full-length GFP. The major band in Coomassie blue staining gel is the full-length GFP. **c** GFP site-specifically installed 4-Azido-Phe is labelled with Cy3-alkyne with Cu(BTTAA)$_2$ as a ligand and ascorbic acid (A.A) as a reducing agent. Wild-type GFP is used as a negative control. **d** Analysis of amber suppression activity of chPheRS-1 by GFP reporter assay with 1 mM of indicated UAAs. **e** Comparison of the incorporation efficiency of 2-Naphthyl-Ala in *E. coli* DH10B cells with chPheRS-1, pylRS[51] and *Mj*.TyrRS[53] (from left to right, $p = 1.56E-4$ and $9.29E-5$). The purified protein yield is shown below the figure. **f** Comparison of the incorporation efficiency of 4-Azido-Phe, and 3-Cyano-Phe with chPheRS-1, pylRS variants[50], and *Mj*.TyrRS[52] (from left to right, $p = 2.19E-8$, $3.40E-7$ and $5.32E-5$). The purified protein yield is shown below the figure. Flow cytometry analysis of amber suppression efficiency of chPheRS-1 (**g**) and chPheRS-3 (**h**) in mammalian cells with or without the addition of 2 mM of indicated UAAs. Error bars represented ±standard deviation of the mean ($n = 3$). Statistical significance is quantified with ordinary one-way ANOVA (***$p < 0.001$). Source data are available in the Source Data file.

previously reported[10,58] (Supplementary Fig. 19A). After selection, a synthetase variant (chPheRS-2) with mutations (Q356N, T467S and A507S), was shown to recognise L-Dopa in *E. coli*. (Supplementary Fig. 19B). The fidelity of L-Dopa incorporation was further confirmed by LC-MS (Supplementary Fig. 19C) and LC-MS/MS analysis (Supplementary Fig. 19D). Fortunately, L-Dopa recognition to the chimeric PheRS/tRNA pair was also functional in mammalian cells (Supplementary Fig. 19E), highlighting the flexibility of the chimeric phenylalanine system. This system could thus be used as a tool to decipher the biological function of histone tyrosine hydroxylation in living cells.

**Incorporating tryptophan analogues.** We noticed that chPheRS-1 recognised endogenous Trp in the absence of UAAs, resulting in a relatively high background activity. (Fig. 5d and Supplementary Fig. 20G) This observation led us to hypothesise that chPheRS can be further engineered to incorporate Trp analogues in prokaryotes and eukaryotes. Although several C5 substituted Trp analogues have previously been site-specifically incorporated using other orthogonalized platforms stated above[28], the introduction of substituted Trp analogues at other positions in the *E. coli*–mammalian shuttle system is still largely elusive. We applied the directed evolution approach to chPheRS[10,54] for Trp

analogues incorporation, similar to L-Dopa recognition described above. We generated a library of synthetase variants by saturated mutagenesis of residues (E391, F464, T467 and A507) that directly interact with the Trp analogues, for positive and negative selections[10,58]. We identified two chPheRS variants (chPheRS-3 carrying the mutations: E391D, T467G and A507G; chPheRS-4 bearing the mutations: F464V, T467G and A507G) showed a strong poly-specificity for a set of Trp analogues. chPheRS-3 preferred the N1 and C6 substituted Trp analogues, while chPheRS-4 exhibited higher activity towards the C7 substituted Trp analogues, with low background activity towards endogenous Trp (Fig. 6a–c). Among these Trp analogues, substitutions such as methyl, chloro and bromo groups were well tolerated with high fidelity of incorporation (Supplementary Fig. 20A–F) and good

yield of purified proteins (Supplementary Table 1). Again, with the 6-Methyl-Trp as an example, the specificity of the evolved chPheRS could be directly applied to mammalian cells without further engineering (Fig. 5h and Supplementary Fig. 18). We noted that the E391D mutation in the synthetase was important for efficient recognition of most Trp analogues, and that mutating E391 to several other amino acids significantly reduced the activity of chPheRS-3 towards Trp analogues (Supplementary Fig. 21). Impressively, the amino acid binding pocket of the chPheRS variants bound to Trp analogues with a larger functional group such as a cyano group. chPheRS-3 recognised 6-Cyano-Trp (6CNW) and chPheRS-4 incorporated 7-Cyano-Trp (7CNW), with high efficiency (Fig. 6b, c and Supplementary Table 1) and fidelity (Fig. 6d, e and Supplementary Fig. 22A-D). These

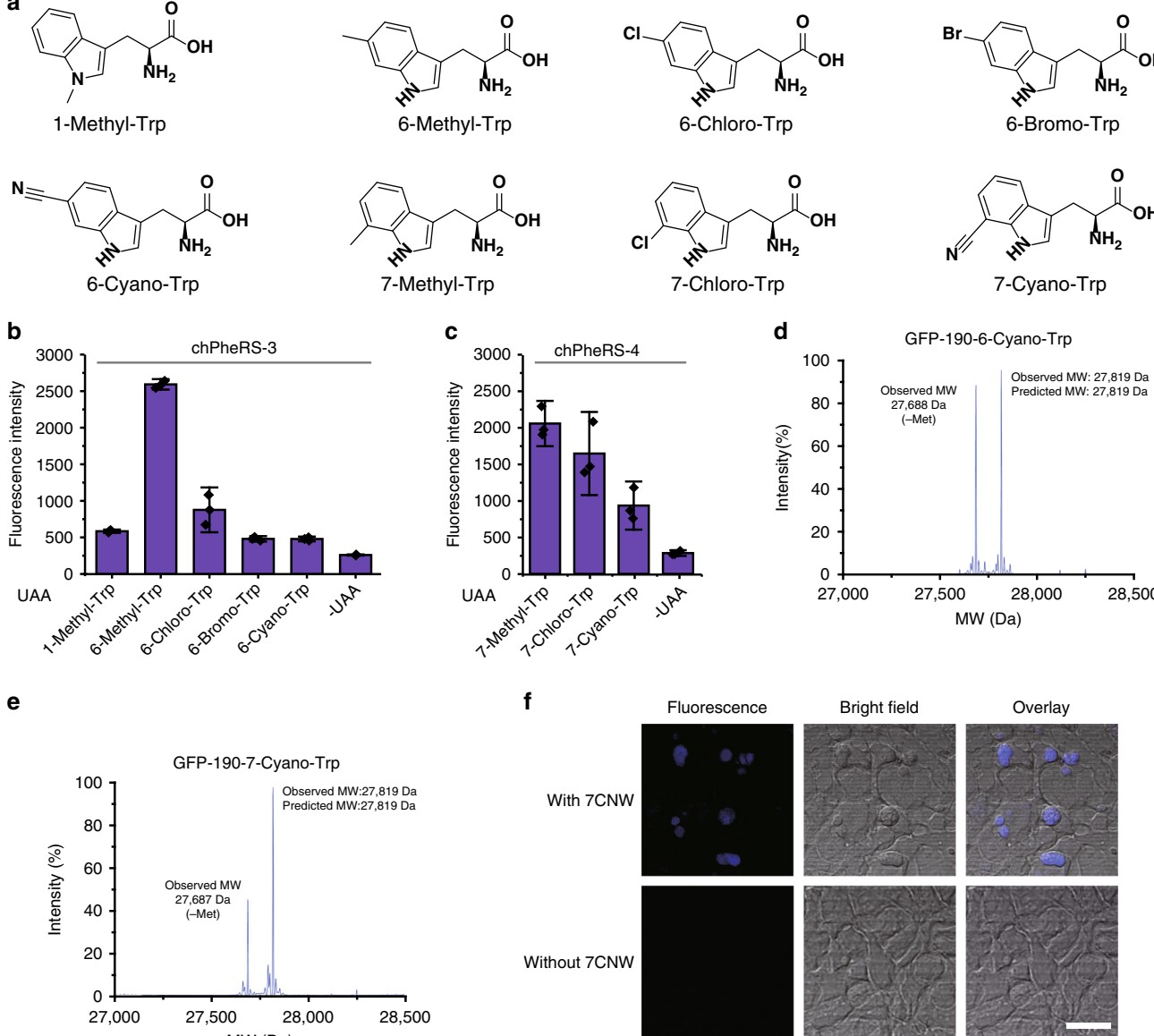

**Fig. 6 Incorporation of Trp analogues with the chimeric PheRS/tRNA pair. a** Chemical structures of the Trp analogues used in this study. Analysis of amber suppression activity of indicated Trp analogues (2 mM) by chPheRS-3 (**b**) and chPheRS-4 (**c**) using GFP reporter assay in *E. coli*. The synthetic details of amino acids 6CNW and 7CNW are described in Supplementary Fig. 23. The chPheRS-3 is a 6CNW recognition synthetase, and the chPheRS-4 is a 7CNW recognition synthetase. Mass spectrometry characterisation of 6CNW (**d**) or 7CNW (**e**) incorporation in GFP. **f** HEK293T cells are co-transfected with two plasmids: one carrying chPheRS-4 gene and chPheT gene, the other bearing Histone H3-E106TAG gene with or without the addition of 2 mM of 7CNW. Transfected cells are then visualised by two-photon microscopy. Scale bar: 25 μm. Error bars represented ±standard deviation of the mean (*n* = 3). Source data are available in the Source Data file.

fluorescent amino acids were chemically synthesised through a four-step procedure (Supplementary Fig. 23). Due to its small size and structural similarity to Trp, the incorporation of Cyano-Trp into proteins has a unique fluorescent property with minimal structural perturbations, which is useful for studying subtle conformation change within the protein and tracking protein localization in cells along with other previously reported fluorescent amino acids[59,60]. Indeed, proteins carrying 6CNW or 7CNW showed a unique fluorescent emission spectrum compared with wild-type (Supplementary Fig. 22E). Histone H3 protein bearing 7CNW at position 106 was clearly visualised in mammalian cells with two-photon excitation microscopy (Fig. 6f). These data demonstrated that a series of Trp analogues with diverse substitutions and chemical properties could be genetically encoded in *E. coli* as well as mammalian cells using our chimera system.

## Discussion

In summary, we have successfully engineered multiple orthogonal aaRS/tRNA pairs by taking full advantage of the key orthogonal components of the pyrrolysine pair. The resulting pairs have comparable activity to that of the pyrrolysine pair and are fully orthogonal in prokaryotes and eukaryotes. Our study broadly expands the number of orthogonal aaRS/tRNA pairs and provides more orthogonal aaRS pockets for genetic code expansion. By further engineering the chPheRS/tRNA pair, we are able to site-specifically incorporate a panel of Phe, Tyr and Trp analogues into proteins in *E. coli* and mammalian cells. Among them, L-Dopa and most of Trp variants (with substitution at N1, C6 and C7) are site-specifically installed into proteins, with fully orthogonal shuttle systems for downstream applications, such as fluorescence imaging in cells. Our work shows that the chPheRS/tRNA pair is a *E. coli*–mammalian shuttle system for genetic code expansion in prokaryotes and eukaryotes. We also expect that engineering of the pockets of generated chimeric synthetases, including chHisRS, chPheRS, chAlaRS and chSerRS, will likely lead to site-specifically incorporate UAAs with distinct structures and applications in mammalian cells. To this end, further engineering of these chimeric synthetases is undergoing in our laboratory to expand its substrate scope and increase its amber suppression efficiency in mammalian cells, as 30 °C is currently observed to be a better working temperature than 37 °C for the incorporation of UAAs in *E. coli*.

In addition, our work indicates that the orthogonality of pyrrolysine system is transplantable with protein and tRNA engineering, and the resulting chimeric system maintains efficiently functional. In the chimera, pylRS TD may act as the primary binder for the recruitment of chimeric tRNAs to its catalytic domain, while the catalytic domain alone exhibits significantly lower activity in our study, and as previously reported[61–64]. This proximity effect of tRNA aminoacylation thus makes chimeras very efficient and orthogonal in vitro and in vivo. Considering that the catalytic domain of most canonical synthetase show much lower activity, this chimeric design can be applied to orthogonalize many, potentially all, aaRS/tRNA pairs. However, the working mechanism of most canonical synthetases is not fully understood and is often very complex, including the formation of multimeric complexes and the allosteric effect between tRNA binding and catalysis. These complexities may make it challenging to apply this strategy to a certain given aaRS/tRNA pair. For example, the catalytic domain of SerRS is reported to form a dimer in vitro[65]. Thus, chSerRS may exist as a dimer between chimeras or even with endogenous SerRS, which explains its lower efficiency after engineering compared to other chimeras. In this scenario, in order to orthogonalize any given aaRS/tRNA pair

with this chimeric design, a better understanding of the working mechanism and additional protein engineering and/or directed evolution will be needed. More broadly, the chimeric aaRSs with transplanted orthogonal components from the pyrrolysine system are still active, which will guide us on transplanting other functional protein in translational machinery through this chimera design.

## Methods

**Reagents.** Unless otherwise noted, all commercial reagents were used without further purification. Primers and genes were synthesised by Tsingke Biotech. Primary Antibodies: Anti-His rabbit polyclonal antibody (Cell Signaling Technology, cat#2365, lot#3), Anti-Flag mouse monoclonal antibody (Sigma-Aldrich, cat#F1804, lot# SLCC6485), Anti-GFP rabbit polyclonal antibody (Cell Signaling Technology, cat#2555, lot#6) and Anti-Alpha-Tubulin rat polyclonal antibody (Santa Cruz Biotechnology, cat#sc-53029, lot#C1517) were used in a dilution of 1:1000. Secondary Antibodies: Goat anti-rat IgG (H + L), HRP conjugate (Proteintech, cat#SA00001-15, lot#20000011), Goat anti-mouse IgG (H + L), HRP conjugate (Proteintech, cat#SA00001-1, lot#20000216) and Goat anti-rabbit IgG (H + L), HRP conjugate (Abmart, cat#M21002, lot#303571) were used in a dilution of 1:5000.

**Instrumentation.** $OD_{600}$ and fluorescent intensity were acquired with Bio Tek Synergy NEO2. Protein gels, including Coomassie blue staining gels and fluorescent gels, and western blotting PVDF membranes, were imaged by Azure Biosystems C400. FACS data was collected by Beckman CytoFlex and processed with FlowJo (LLC). LC-MS analysis was performed on an Xevo G2-XS QTOF MS System (Waters Corporation). LC-MS/MS analysis was performed on a Q Exactive Orbitrap mass spectrometer in conjunction with a Proxeon Easy-nLC II HPLC (Thermo Fisher Scientific) and Proxeon nanospray source.

**Cell culture procedures.** HEK 293T cells (ATCC) were maintained in an exponential growth as a monolayer in Dulbecco's Modified Eagle Medium (DMEM, Thermo Fisher Scientific), high glucose, 10% foetal bovine serum (FBS, Thermo Fisher Scientific), 1% penicillin–streptomycin and incubated at 37 °C in 5% $CO_2$.

**Assessment of amber suppression efficiency.** The plasmid pNEG bearing the GFP-190TAG and a chimeric tRNA and the plasmid pBK carrying the corresponding chimeric synthetase were co-transformed into chemically competent DH10B cells. The transformed cells were recovered in 2xYT medium for 1 h with shaking at 37 °C and plated on LB agar containing 50 μg/ml kanamycin and 100 μg/ml ampicillin for 12 h at 37 °C. At the same time, the cells were transformed with pNEG vector alone were used as a negative control. A single colony was picked and grown in 2 ml of 2xYT medium containing required antibiotics at 37 °C until $OD_{600}$ reaching ~0.8. The protein expression was induced by the addition of arabinose with a final concentration of 0.2% at 30 °C for 16 h with or without the addition of the corresponding unnatural amino acids. After induction, 1 ml of cell cultures were collected by centrifuging, and then lyzed by 150 μl BugBuster Protein Extraction Reagent (Millipore) for 20 min at room temperature. The supernatant of the lysate (100 μl) was transferred to a 96-well cell culture plate (Costar). And GFP signals of the supernatant were recorded by Bio Tek Synergy NEO2 with a background subtraction and normalised by the bacterial density ($OD_{600}$) that was measured by Bio Tek Synergy NEO2 as well. Assessment of amber suppression efficiency of the pylRS system was measured by the same procedure with or without the addition of 2 mM Boc-L-lysine (BocK) into the medium.

**GFP expression, purification and LC-MS analysis.** For GFP expression and purification, overnight cultured DH10B cells were diluted 1:100 into 100 ml of fresh LB medium that was supplemented with required antibiotics. The cells were grown until $OD_{600}$ reaching ~0.8. L-arabinose was added with a final concentration of 0.2% to induce the expression of GFP (30 °C, 220 rpm, 16 h) with or without the addition of the corresponding unnatural amino acids. The cells were harvested by centrifuging at 3000 *g* for 5 min at 4 °C. The resulting cell pellets were suspended in ice-cold NTA-0 buffer (25 mM Tris, 250 mM NaCl, pH 8.0), and sonicated. The suspension was centrifuged at 10,000 *g* for 60 min at 4 °C. The resulting supernatant was purified via $Ni^{2+}$-affinity chromatography on chelating Sepharose equilibrated with NTA-0 buffer, and washed with six volumes of NTA-0 buffer containing 50 mM imidazole. The proteins were eluted with NTA-0 buffer supplemented with 500 mM imidazole. Purified proteins were subjected to SDS-PAGE and LC-MS analysis. Purified protein yields were obtained by measuring the absorbance of GFP variants at 395 nm using a spectrometer, and the data were summarised in Supplementary Table 1.

For LC-MS analysis, the purified proteins were analysed on an Xevo G2-XS QTOF MS System (Waters Corporation) equipped with an electrospray ionisation source in conjunction with Waters ACQUITY UPLC I-Class plus. Separation and desalting were carried out on a Waters ACQUITY UPLC Protein BEH C4 Column (300 Å, 2.1 × 50 mm, 1.7 μm). Mobile phase A was 0.1% formic acid in water and

mobile phase B was acetonitrile with 0.1% formic acid. A constant flow rate of 0.2 ml/min was used. Data were analysed using Waters UNIFI software. Mass spectral deconvolution was performed using UNIFI software (version 1.9.4, Waters Corporation). The molecular weight of the protein was predicted using the ExPASy Compute pI/Mw tool, and chromophore maturation in GFP was also considered in the calculation.

**Assessment of amber suppression efficiency in mammalian cells**. The chimeric synthetases and tRNA genes were PCR-amplified and inserted into pcDNA3.1 vector by Gibson Assembly under the control of CMV and U6 promoter, respectively. The vector carried the chimeric tRNA alone was made as a negative control. HEK 293T cells were grown in DMEM medium supplemented with 10% foetal bovine serum and 1% penicillin–streptomycin. Cells were co-transfected with the pcDNA3.1 vector and the pEGFP-mCherry-T2A-EGFP-190TAG at ratio 1:1 (μg:μg). Transfections were performed by lip2000 reagent (BioSharp) according to the manufacturer's protocol with or without the addition of the corresponding amino acids. Imaging was performed 48 h after transfection. Live HEK 293T cells were imaged on IX71 Inverted Microscope equipped with a 10× objective lens (UPlanSApo, Olympus) at FITC channel. All images were analysed and processed with ImageJ software (National Institutes of Health).

**Selection of the chPheRS variants for UAAs in E. coli**. Selections of active chimeric PheRS variants for UAAs were carried out with one round of negative selection and subsequently positive selection, using the following libraries: L-Dopa (Q356NNK, T467NNK and A507NNK) and Trp analogues (E391NNK, F464NNK, S470NNK, C487NNK, T467G and A507G). Briefly, chPheRS libraries in pBK vectors were firstly electroporated into DH10B competent cells harbouring the negative selection plasmid pNEG-Barnase-Q3TAG-D45TAG-chPheT. After the negative selection, the surviving pool was then electroporated into DH10B competent cells harbouring the positive selection plasmid pNEG-CAT112TAG-chPheT-GFP190TAG that contains CAT and GFP dual reporter genes with an amber codon (112TAG for CAT and 190TAG for GFP), respectively. The transformed cells were recovered for 4 h at 37 °C then plated on LB agar containing 50 μg/ml kanamycin, 100 μg/ml ampicillin, 10 μg/ml chloramphenicol and 0.2% L-arabinose in the presence of 2 mM desired UAAs for 12 h at 37 °C and subsequently at 30 °C for 48 h. The clones with fluorescence in the plate were picked and expressed in the presence or absence of 2 mM desired UAAs. The fluorescence measurements were carried out as described above. Ten clones with the most prominent UAA-dependent GFP fluorescence were sent to sequencing.

**Two-photon fluorescence imaging**. For two-photon fluorescence imaging with 7CNW in cells, HEK 293T cells were grown on glass coverslips (801007, NEST) and co-transfected the pcDNA3.1 vector carrying the chPheRS-4/chPheT pair with pEGFP-H3-E106TAG vector. After 6 h of transfection, the medium was replaced with fresh medium containing 2 mM 7CNW, and the cells were cultured for further 36 h. Subsequently, cells were washed with PBS buffer for three times, fixed by 2% PFA for 10 min and mounted on a glass slide using Fluoroshield ab104135 mounting medium (Abcam) for imaging. Two-photon fluorescence imaging was performed with Olympus FVMPE-RS multiphoton imaging system equipped with a 25 × 1.05NA water immersion objective lens at room temperature. The cells were excited at 690 nm (two-photon) to acquire 7CNW fluorescence images at 410–460 nm. The imaging setup was controlled by Olympus FV31S-SW. All images were analysed and processed with ImageJ software (National Institutes of Health).

**FACS analysis of live cells**. For FACS analysis of live cells, HEK 293T cells were grown in six-well plates (Corning) and co-transfected the pcDNA3.1 vector bearing the aaRS/tRNA pair (pcDNA vector bearing only chimeric tRNA as a negative control) and pEGFP-mCherry-T2A-GFP190TAG-His$_6$ at ratio 1:1 (μg:μg). Transfection was performed using lip2000 reagent (BioSharp) according to the manufacture's protocol with or without the addition of the corresponding amino acids. After transfection (48 h), cells were trypsinized and taken up in full medium before centrifugation. Cells were centrifuged at 1400 × g for 3 min, washed, and resuspended in PBS. FACS instrument (Beckman CytoFlex) was set up according to the manufacturer's instructions. HEK 293T cells were used to set appropriate forward scatter and side scatter gains. The fluorescent protein expressed cells were used to set FITC and PE gains and gate. At least 50,000 single cells were analysed per condition. Lastly, GFP fluorescence was acquired at the FITC channel, and mCherry fluorescence was acquired at the PE channel. FACS data were analysed and processed with FlowJo (LLC). Flow cytometry gating strategy is shown in Supplementary Fig. 8A.

**Cloning, expression and purification of synthetases**. The chimeric synthetases and canonical synthetases were amplified and then inserted into the pET28a vector by Gibson assembly. The expression plasmids were transformed into chemically competent E. coli BL21 (DE3) cells and plated onto LB agar containing 50 μg/ml kanamycin. A single colony was picked and grown in 400 ml of LB medium containing 50 μg/ml kanamycin at 37 °C until OD$_{600}$ reaching ~0.8. The protein expression was induced by the addition of isopropyl β-D-thiogalactoside with a

final concentration of 1 mM at 16 °C for 16 h. After induction, the cells were harvested by centrifuging at 3000 g for 20 min at 4 °C. The resulting cell pellets were suspended in ice-cold NTA-0 buffer (50 mM Tris-Cl, 500 mM NaCl, pH 7.4), and sonicated. The suspension was centrifuged at 10,000 g for 60 min at 4 °C. The resulting supernatant was purified via Ni$^{2+}$-affinity chromatography on chelating Sepharose equilibrated with NTA-0 buffer, and washing with NTA-0 buffer containing 50 mM imidazole. The proteins were eluted with NTA-0 buffer supplemented with 500 mM imidazole and dialysed against NTA-0 buffer containing 1 mM DTT. Protein concentrations were determined by absorbance at 280 nm, and the extinction coefficient of each protein was predicted with the ProtParam tool (https://web.expasy.org/protparam/).

**The extraction and purification of total tRNAs**. To extract total tRNAs and total tRNAs containing chimeric tRNA from E. coli[66], DH10B cells and DH10B cells transformed with the plasmid pNEG bearing the GFP-190TAG and a chimeric tRNA were cultured at 37 °C in LB medium containing required antibiotics. The cells were harvested at mid-log phase by centrifuging at 3000 g for 5 min at 4 °C and washed with water once. The cells were resuspended in 1.5 volumes of buffer A (1.0 mM Tris–HCl, 10 mM MgCl$_2$, pH 7.2) and lysed with 1.5 volumes of phenol saturated with water. After mixing in head-over-shaking (15 rpm) at 4 °C for 1 h, the phases were separated by centrifuging at 12,000 g for 20 min at 4 °C. The aqueous phase was further purified with one volume of chloroform. The aqueous phase containing total tRNAs were deacylated by 300 mM NaOH (40 μl for 680 μl aqueous phase) and incubated at 42 °C for 1 h. The aqueous phase was neutralised by 40 μl NaOAc (3 M, pH 5) and precipitated with 2.3 volumes of absolute ethanol at 4 °C for 1 h. The deacylated total tRNAs were harvested by centrifuging at 12,000 g for 20 min at 4 °C and dissolved with DEPC-treated water.

To extracted total tRNAs from HEK 293T cells[67], the harvested cells were resuspended with buffer B (50 mM NaOAc, 150 mM NaCl, 10 mM MgCl$_2$, 0.1 mM EDTA, pH 5). The following steps were the same as tRNA extraction from E. coli.

**In vitro aminoacylation assay**. In vitro aminoacylation assay was conducted according to the previous papers[67,68]. For chimeric Phe and Ala system, 100 nM synthetases were incubated with 0, 30 μg of extracted tRNAs in reaction buffer (50 mM Tris–HCl, 10 mM MgCl$_2$, 20 mM KCl, 2 mM DTT, pH 7.4) supplemented with 4 mM ATP, 0.1 mg/ml BSA and 80 μM L-phenylalanine, 17.8 μM L-[$^{14}$C]-phenylalanine for chimeric Phe system or 50 μM L-alanine, 33.3 μM L-[$^{14}$C]-alanine for chimeric Ala system at 37 °C for 40 min. After incubation, the aliquots were spotted onto Whatman GF/C glass microfiber filter pre-equilibrated with 10% cold TCA. The paper filters were washed with cold 10% TCA four times and then 95% ethanol four times. After washing, the filter papers were visualised by the liquid scintillation counter to quantify the radioactivity. For chimeric His system, 25 nM synthetase was incubated with 0, 10 and 30 μg of extracted tRNAs in reaction buffer (50 mM Tris–HCl, 10 mM MgCl$_2$, 20 mM KCl, 2 mM DTT, pH 7.4) supplemented with 0.1 mM ATP, 1 mM L-histidine and 0.1 mg/ml BSA at 37 °C for 2 h. The aminoacylation was measured through the production of AMP with AMP-Glo assay (Promega) according to the manufacturer's instructions.

**Reporting summary**. Further information on research design is available in the Nature Research Reporting Summary linked to this article.

## Data availability

Any Supplementary Information (methods and figures), Supplementary Data (DNA sequences and protein sequences), and chemical compound information are available in this paper. LC-MS/MS data underlying Supplementary Figs 3C, 13E, 15G, 16D, 19D, and 22C,D have been deposited to the ProteomeXchange Consortium via the PRIDE[69] partner repository with the dataset identifier PXD018660 [https://www.ebi.ac.uk/pride/archive/projects/PXD018660]. The source data underlying Figs. 2d, 3b–e, 4a–g, 5c–f, 6b–f, and Supplementary Figs 6B, 7, 10B, 13A, 14, 15B,C, 16B,C, 17C–E, 19B,C, 20A–G, 21 are provided as a Source Data file. The uncropped image of native gels (Figs. 3e, 4b, e, and Supplementary Fig. 6B) are provided in Supplementary Figs. 3D, 13B, 15D, and Source Data-Supplementary 6B, respectively. All relevant data are available from the corresponding author upon reasonable request. Source data are provided with this paper.

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

## Acknowledgements

We thank the National Natural Science Foundation of China (Grant 91953113), the National Key R&D Program of China (Grant 2019YFA09006600), the Fundamental Research Funds for the Zhejiang Provincial Universities (Grant 2019XZZX003-19, 2018QN81011), and China Postdoctoral Science Foundation (2019M652072 for W. D.) for financial support. We are grateful to the core facility of Life Sciences Institute, Prof. Jie P. Li for supplying equipment for LC-MS analysis, Dr. Hangjun Wu (the centre of Cryo-Electron Microscopy) for his technical assistance, Dr. Shitang Huang for helping with the radiolabelling assay, and Prof. Xing Guo for mammalian cell lines. And we gratefully thank Prof. Xiangwei He and Dr. Vivian Y. Yu for their helpful discussions.

## Author contributions

S.L. conceived the idea and supervised the study. W.D. designed the chimera strategy. W.D. and H.Z. conducted most experiments and analysed the data together. Y.C., Y.Y., J.Z. and J.W. provided technical advice, assisted with MS data analysis and interpretation. B.Z. synthesised tryptophan analogues. S.L., W.D. and H.Z. wrote the manuscript. All authors commented on the final draft of the manuscript.

## Competing interests

The authors declare the following competing interests: S.L., W.D., H.Z. and Zhejiang University have filed two patent applications (Chinese patent application number: 201910440254.8 and 201911095377.9) describing the chimeric design for orthogonalizing aaRS/tRNA pairs and its application in genetic code expansion. All other authors declare no competing interests.
