## [Peer Review File · Nature Communications]

Reviewers' Comments:

Reviewer #1:

Remarks to the Author:

The re-revised manuscript from Ding et. al has addressed many of the concerns that reviewers had but requires some additional modifications before publication. The manuscript has a better flow and the ideas are more clearly laid out than previous versions. However, the grammar still requires some work to make it easier to read and clarify the overall results.

1. In the abstract it should clearly state that the PyIRS system is currently the ideal system for GCE in both eukaryotes and prokaryotes. The way it is currently structured does not come across that way and could be argued that MjTyrRS is also an ideal system for GCE in prokaryotes.
2. The authors claim that by simply transplanting the key orthogonal components from the PyIRS system into an aaRS/tRNA system you get fully orthogonal and high activity. This is a bit of an overstatement as the manuscript goes through major detail on how further engineering of the aaRS and tRNA is required in most cases (lines 88-92).
3. The labels on Figure 3F are not consistent, one discusses aaRS and amino acid and the other is aaRS and tRNA. It is not clear in the figure or its legend which amino acid is used for chHisRS and presuming that tRNAPyl is used with PyIRS.

Reviewer #2:

Remarks to the Author:

Since I am not one of the original reviewers. All my comments are based on what authors have been put together to address previous reviewers' concerns. The authors have provided point-by-point responses to the previous concerns. Although most of the previous concerns have been addressed, there are still some issues left.

Reviewer 1

Point 1:

It is hard to judge from Figure S16B on whether or not 4-azido-Phe was incorporated into the proteome. The exposure has been tuned too low so that even markers (assumed for Western blotting) cannot be observed. Proper adjustment of exposure is needed for showing both background incorporation and a positive control.

Point 2:

Concerns have been addressed.

Point 3:

Concerns are addressed.

Point 4:

Concerns are addressed.

Point 5:

Concerns are partially addressed. All integrated LC-MS spectra have quite broad bands. It is no way to judge from these spectra that the incorporation is orthogonal. High resolution MS are needed.

Pont 6:

Concerns are addressed.

Point 7:
Concerns are addressed.

Reviewer 2

Point 1:
The title has been changed according to the reviewer's suggestion.

Point 2:
Additional data are provided to show 30 degree is optimal. This will be fine for E. coli but highly problematic for mammalian cells. Since this is what it is, the authors need to point out in the paper that the system will have some disadvantage when used in mammalian systems.

Point 3:
Concerns are addressed.

Reviewer 3

Point 1:
Concerns are addressed.

Point 2:
Concerns are addressed.

Point 3:
concerns are addressed.

Point 4:
Concerns are addressed.

Point 5:
Concerns are addressed.

Point 6:
Concerns are addressed.

Point 7:
Concerns are addressed.

Point 8:
Concerns are addressed.

Point 9:
Concerns are partially addressed. Figures 5D, 5G, and S13 need to be redrawn. There are only one data points with multiple repeats. It is misleading (more leaning to not right) to draw a count vs ug chart.

Point 10:
Concerns have been addressed.

Point 11:
Concerns have been addressed.

Point 12:
No comments.

Reviewer #1:

The re-revised manuscript from Ding et. al has addressed many of the concerns that reviewers had but requires some additional modifications before publication. The manuscript has a better flow and the ideas are more clearly laid out than previous versions. However, the grammar still requires some work to make it easier to read and clarify the overall results.

We really appreciate the referee for revising our paper and providing insightful comments. We have proofread the grammar of our manuscript with major changes highlighted in red.

1. In the abstract it should clearly state that the PylRS system is currently the ideal system for GCE in both eukaryotes and prokaryotes. The way it is currently structured does not come across that way and could be argued that MjTyrRS is also an ideal system for GCE in prokaryotes.

Thank you for pointing this out. We re-wrote this sentence to make a clearer statement (page 1, text in red).

2. The authors claim that by simply transplanting the key orthogonal components from the PylRS system into an aaRS/tRNA system you get fully orthogonal and high activity. This is a bit of an overstatement as the manuscript goes through major detail on how further engineering of the aaRS and tRNA is required in most cases (lines 88-92).

We have toned down the statement according to the referee's suggestion (page 4, text in red).

3. The labels on Figure 3F are not consistent, one discusses aaRS and amino acid and the other is aaRS and tRNA. It is not clear in the figure or its legend which amino acid is used for chHisRS and presuming that tRNAPyl is used with PylRS.

Thanks for pointing it out. We have corrected the labels to make it consistent (Figure 3F), and edited its legend to clarify the result (page 19, text in red).

Reviewer #2:

1. It is hard to judge from Figure S16B on whether or not 4-azido-Phe was incorporated into the proteome. The exposure has been tuned too low so that even markers (assumed for Western blotting) cannot be observed. Proper adjustment of exposure is needed for showing both background incorporation and a positive control.

We thank the referee for the suggestion. We have conducted the in-gel fluorescence imaging assay again for the cell lysate samples with purified GFP-190-4-Azido-Phe protein as a positive control. And we have provided the fluorescent images with both long exposure time and short exposure time in Figure S16B.

2. Concerns are partially addressed. All integrated LC-MS spectra have quite broad bands. It is no way to judge from these spectra that the incorporation is orthogonal. High resolution MS are needed.

We have now re-run all our protein samples containing both natural amino acids and unnatural amino acids with a high-resolution MS (ESI-TOF MS). In the high-resolution LC-MS spectra, the major monoisotopic peak and its minor isotope peaks are clearly detected. We have updated the LC-MS data in Figures 3D, 4C, 4F, 6D-E, S2B, S12D, S14F, S15C, S16C-E, S18C, S19A-G, and S21A-B for evaluation.

3. Additional data are provided to show 30 degree is optimal. This will be fine for E. coli but highly problematic for mammalian cells. Since this is what it is, the authors need to point out in the paper that the system will have some disadvantage when used in mammalian systems.

We have added this point in the conclusion section (Page 15, text in red). Thanks for the suggestion.

4. Concerns are partially addressed. Figures 5D, 5G, and S13 need to be redrawn. There are only one data points with multiple repeats. It is misleading (more leaning to not right) to draw a count vs ug chart.

We have re-drawn the figures to clarify the results (Figure 5D, 5G, and S13). Thanks for pointing it out.

Reviewers' Comments:

Reviewer #2:

Remarks to the Author:

All concerns from this reviewer have been addressed. It is ready for acceptance.